# Study of the Potential of Water Treatment Sludges in the Removal of Emerging Pollutants

**DOI:** 10.3390/molecules26041010

**Published:** 2021-02-14

**Authors:** Rita Dias, Diogo Sousa, Maria Bernardo, Inês Matos, Isabel Fonseca, Vitor Vale Cardoso, Rui Neves Carneiro, Sofia Silva, Pedro Fontes, Michiel A. Daam, Rita Maurício

**Affiliations:** 1CENSE—Center for Environmental and Sustainability Research, School of Science and Technology, NOVA University Lisbon, 2829-516 Caparica, Portugal; db.sousa@campus.fct.unl.pt (D.S.); m.daam@fct.unl.pt (M.A.D.); rmr@fct.unl.pt (R.M.); 2LAQV/REQUIMTE, School of Science and Technology, NOVA University Lisbon, 2829-516 Caparica, Portugal; maria.b@fct.unl.pt (M.B.); ines.matos@fct.unl.pt (I.M.); blo@fct.unl.pt (I.F.); 3EPAL—Empresa Pública de Águas Lives S.A., AdP—Grupo Águas de Portugal, 31700-421 Lisboa, Portugal; vitorcar@ADP.PT (V.V.C.); rcarnei@ADP.PT (R.N.C.); sofia.silva-e@ADP.PT (S.S.); p.fontes@ADP.PT (P.F.)

**Keywords:** emerging pollutants, water treatment sludges, adsorption processes, circular economy

## Abstract

Presently, water quantity and quality problems persist both in developed and developing countries, and concerns have been raised about the presence of emerging pollutants (EPs) in water. The circular economy provides ways of achieving sustainable resource management that can be implemented in the water sector, such as the reuse of drinking water treatment sludges (WTSs). This study evaluated the potential of WTS containing a high concentration of activated carbon for the removal of two EPs: the steroid hormones 17β-estradiol (E2) and 17α-ethinylestradiol (EE2). To this end, WTSs from two Portuguese water treatment plants (WTPs) were characterised and tested for their hormone adsorbance potential. Both WTSs showed a promising adsorption potential for the two hormones studied due to their textural and chemical properties. For WTS1, the final concentration for both hormones was lower than the limit of quantification (LOQ). As for WTS2, the results for E2 removal were similar to WTS1, although for EE2, the removal efficiency was lower (around 50%). The overall results indicate that this method may lead to new ways of using this erstwhile residue as a possible adsorbent material for the removal of several EPs present in wastewaters or other matrixes, and as such contributing to the achievement of Sustainable Development Goals (SDG) targets.

## 1. Introduction

One of the most prominent problems affecting the world’s population is insufficient access to clean water and sanitation. According to Sustainable Development Goal (SDG) 6 “clean water and sanitation” of the Synthesis Report on Water and Sanitation [1], over two billion people are living in countries that are experiencing high water stress conditions. This situation is a result of an overuse of water resources with significant impacts on their sustainability. Concurrently, water quality problems persist in water bodies of both developed and developing countries, such as the loss of the pristine quality conditions, changes in hydromorphological characteristics and an increase in concentrations of emerging pollutants (EPs) [1]. The clearest link between water and the circular economy is to consider drinking water and wastewater treatment plants as resource recovery installations, stimulating the recovery and valorisation of treated water and wastewater materials [2]. The circular economy concept provides ways of advancing towards sustainable water resource management that can be implemented in the water sector to achieve circularity between drinking water and wastewater resources [3].

Conventional wastewater treatment plants (WWTPs) are not entirely effective in the removal of EPs from wastewater (WW) since they were conceptually designed for the removal of macropollutants such as nutrients, suspended solids, pathogenic microorganisms and trace elements. Therefore, EPs such as pharmaceuticals and endocrine disrupting compounds (EDCs) may go through the treatment system unchanged or are only partially removed, leading to their detection in WW-receiving water bodies and WWTP discharges in concentrations ranging from ng/L to mg/L [4,5,6,7,8]. EDCs may lead to the modification of the natural function of the endocrine system in wildlife by (i) blocking or copying the normal effect of hormones; (ii) affecting their synthesis or metabolism; and (iii) changing hormone receptor levels [8,9,10,11]. Among EDCs, estrogens such as 17β-estradiol (E2) and 17α-ethinylestradiol (EE2) have often been indicated as particularly problematic compounds with high associated risks [12,13,14]. E2 is a natural steroid hormone, which is secreted by humans and animals. The synthetic steroid hormone EE2 is based on the natural estrogen E2, which is used in oral contraceptives and hormone replacement therapies. These estrogens are largely excreted by humans and animals through urine and faeces and end up in the environment mainly through the discharge of WWTP effluents and the disposal of animal waste [10,15,16]. Secondary treatment is not effective in the removal of both of these compounds, especially for EE2 due to its recalcitrant nature. As such, it is known that WWTP final discharges are the main source of both E2 and EE2 in the aquatic environment. These chemicals have the potential to bioaccumulate and enter the food chain, posing ecotoxicity to aquatic organisms and implying risks to aquatic ecosystems. Several studies have reported the impact of E2 and EE2 on fish life, such as feminising male fish, reducing testicle size, reducing reproductive fitness, lowering sperm count, inducing the reproduction of vitellogenin, altering other reproductive characteristics and causing behavioural changes [9,17,18]. The uptake by fish and the presence of these compounds in the raw water that is used to produce drinking water to supply human communities also have impacts on human life and have previously been reported. The consumption of these compounds at concentrations above the safety thresholds can increase the risk of cancer and induce cardiovascular diseases [9,16,17].

Several technical solutions for EP removal have previously been developed that allow for their integration with existing treatment processes in an expedient way [7,19]. However, most of the methods are not techno-economically viable for large-scale implementation, and thus, comprehensive research is necessary to develop suitable, low cost, eco-friendly and efficient technologies to remove different kinds of EPs from WW [8,19,20,21,22]. The adsorption process with activated carbon (AC) is considered by many authors to be one of the most promising treatment processes with high EP removal capacity, mainly because (i) it is simple to design and operate; (ii) it has a low investment cost; (iii) it allows reuse and regeneration; and (iv) it does not generate toxic by-products [8,19,21,22,23,24]. Several studies have also already demonstrated that pure AC is able to effectively remove and lower the toxicity of E2 and EE2 in distilled water, drinking water and WW [25,26,27,28,29,30,31,32,33,34,35]. For example, Gökçe and Arayici [33] obtained a removal rate for E2 of 88% with AC produced from sewage sludge. In this referred work [33], the sludges were modified and submitted to several procedures in order to obtain sludge-based adsorbents.

Sludge-based adsorbents have been reported in the literature for the removal of several pollutants from water treatment plants (WTPs) and WWTPs [36,37]. Research efforts, however, have focused on their removal efficacy for compounds like heavy metals [38,39,40,41,42,43], dyes [44,45,46], phenols [47,48], phosphorus and phosphate [49,50,51,52] and antibiotics [53]. Consequently, the adsorbent potential of sludge for compounds like E2 and EE2 remains unknown. Clara et al. [54] tested the capacity of sludge from WWTP (without AC content) to adsorb E2 and EE2 in the mg/L range and noted a high adsorption potential of the tested sludge.

Drinking water treatment plant sludges (WTSs) have been recycled as aggregates, soil improvement agents and environmental remediation materials. The use of WTSs as an adsorbent in WW treatment is related to their high concentration of amorphous aluminium and ferric ions. These ions have a high affinity to phosphors and heavy metals through ion-exchange and complexation mechanisms [55,56,57,58]. The first report indicating the potential use of WTS containing AC was made by Lee et al. [58]. They evaluated the possibility of regenerating the AC and coagulants present in WTS via pyrolysis to produce multifunctional remediation material for the removal of pollutants present in WW.

Considering (i) the capacity of AC to remove EPs in an adsorbent process and (ii) the possibility to reuse WTS for several applications, it is envisaged that there is a potential use of WTS containing AC for the removal of EPs. As detailed above, however, current knowledge in this field remains limited. The main goal of this study was therefore to evaluate the potential of two unmodified WTSs with high content of activated carbon, without reactivation, as an adsorbent for the removal of two selected EPs, namely, E2 and EE2. No additional AC was incorporated in the sludges used nor was any kind of sludge treatment/modification made (including an AC reactivation step). Consequently, the AC content in the test sludges only had its origin from the conventional liquid treatment phase in the drinking WTP process. The elemental and mineral composition, ash content, pH at the point of zero charge (pHpzc), thermogravimetric analysis and textural characterisation of the sludges were determined. The removal potential of the two hormones by the sludges was also assessed, allowing an evaluation of the circularity of this residue and thus the possibility of transforming it into a new adsorbent material.

## 2. Results and Discussion

### 2.1. Water Treatment Sludge Characterisation

The elemental analysis, ash content, pHpzc and textural parameters of the two WTSs are presented in Table 1. WTS samples presented high ash contents: 42.9% for WTS1 and 30.9% for WTS2, which was expected given their origin. Nevertheless, these values are slightly lower than those observed by Lee et al. [58], who reported an ash content of around 51.0% for sludge obtained in South Korea. The variation between the ash content of these WTSs could be explained by the different reagents used in the treatment line and therefore the different mineral content. In fact, the presence of some mineral elements could be important because they may promote the formation of strong interactions with organic pollutants and as such promote their removal from water [58,59,60].

The higher content of carbon observed in WTS2 as compared to WTS1 indicates that WTS2 may have a higher incorporation of activated carbon in its composition. The higher carbon content is reflected in the lower ash content of WTS2 and consequently in a higher surface area and porosity (Table 1). On the other hand, since WTS1 presented a higher ash content, it has a lower surface area given that the ash may block the pores in the sludge [58,61]. The pHpzc is directly related to the ash content of the samples: WTS1 is the more alkaline sample because it presented the higher mineral content. WTS2, on other hand, had a more neutral pHpzc, which indicates that WTS2 is more similar to alum sludge [57,62]. According to these results, WTS1 and WTS2 surfaces were positively charged in water solutions with a pH of below 11.29 and 7.46, respectively (Table 1).

The results from the mineral analysis (Table 2) demonstrate that the elements present in higher concentrations in WTS2 were aluminium (Al; 5.4%) and phosphorous (P; 3.5%), followed by iron (Fe; 1%) and calcium (Ca; 0.6%), as visualised in Figure 1. For WTS1, the elements with the highest concentrations were calcium (Ca; 11.5%), followed by aluminium (Al; 5.3%), phosphorous (P; 3.4%) and iron (Fe; 0.7%) (Table 2; Figure 1).

The main difference in mineral composition between the two WTSs was related to the calcium content (Table 2), which could be explained by the addition of insoluble limestone residues in the final stage of the sludge treatment of WTS1. To understand the thermal behaviour of WTS during an activation/regeneration process, it is fundamental to study their thermal decomposition through thermogravimetric analysis (TGA), which is presented in Figure 2.

During the TGA analysis, the total mass loss was 34% and 27% for WTS1 and WTS2, respectively. These values may suggest that, through a regeneration/activation process, a significant mass was lost [61]. Both samples presented weight loss associated with water loss at 105 °C: WTS2 humidity was 6.5%, while for WTS1, this was 2.5%. WTS1 also presented a significant mass loss above 700 °C, which could be related to the decomposition of CaCO_3_ [59,63]. In fact, WTS1 was enriched with calcium that was probably present as carbonates.

### 2.2. Removal of Emerging Pollutants from Water

The removal efficiency experiment of the adsorbates with the WTS adsorbents was performed by using a dosage of 5 g WTS/L. This dosage was based on the concentrations used by Lee et al. [58] in their adsorption experiment, which were 4, 5 and 20 g/L.

The overall removal efficiency for both hormones was above 50% in both WTSs (Table 3). These results are in line with the removal efficiencies reported in the studies by Yoon et al. [30], Fuerhacker et al. [31], Gökçe and Arayici [33] and Rowsell et al. [32]. It should be noted, however, that these previous studies were performed using pure AC, with higher surface areas and pore volumes. Thus, few studies have been conducted to date using raw (unmodified) WTS (with or without an activation process) for the removal of EPs, as done in the present study. Evidently, such studies would be useful to meet the SDG targets and circular economy commitments.

The removal efficiency in WTS2 was over 90% (for both E2 and EE2) at any of the initial hormone concentrations tested (Table 3). This result is comparable to the one presented by Yoon et al. [30] using virgin powdered activated carbon (PAC), who achieved a removal efficacy of 99% for both E2 and EE2 with a contact time of 24 h. Ifelebuegu et al. [34] also achieved a 96% removal efficacy of EE2 with 0.1 g/L wood-based granular activated carbon (GAC). Regarding sludge-based carbons, Clara et al. [54] used activated and raw sludge from a WWTP to evaluate the adsorption capacity of E2 and EE2. These authors concluded that both compounds showed a high adsorption affinity to the adsorbent, and within a contact time of 24 h, no difference between activated and inactivated sludge was detected. Some authors have previously suggested that the octanol/water partition coefficient (Log Kow) values for estrogenic compounds, which vary between 2.5 and 4.0, can roughly predict the sorption behaviour. E2 and EE2 are often cited as moderately hydrophobic and to have a tendency to adsorb to the solid phase [9,30,32]. These properties could thus indicate an easy interaction with the adsorbent [35]. The pH value of the test matrix is also an important factor that affects the adsorption capacity of sludge as an adsorbent [37]. In this preliminary study, the pH of the solution with distilled water and hormones was around 3. Previous studies have demonstrated that adsorption is higher at acidic pH and that the highest adsorption of E2 and EE2 occurs at neutral pH (6-7) [25]. Both E2 and EE2 are protonated (with pKa 10.4 and 10.7, respectively), while, according to the WTS pHpzc values, WTS surfaces are positively charged, which may support a positive interaction (no electrostatic repulsion) between adsorbent and adsorbate. Besides the influence of the AC properties on its adsorption capacity, the presence of other elements could also have influenced the adsorption performance of the sludges tested. For example, the high aluminium content of the sludges (Table 2) may have influenced the adsorption process, although there appears little consensus in the literature on this influence [36,38,51,62]. On the one hand, aluminium has been indicated to improve the adsorption capacity of WTS (e.g., Lee et al. [58]). On the other hand, other studies have concluded that aluminium as a coagulant [64,65] in WTS has only a minimal impact on the removal efficacy of steroid compounds [66]. Therefore, it is of utmost importance that further research is conducted to assess the role of aluminium on the adsorption capacity of WTSs containing AC for these hormones.

WTS2 had a higher removal efficiency when compared with WTS1 since the final concentration was lower than the limit of quantification (LOQ) (Table 3), which could be related to a number of factors. The most evident reason is the higher surface area and pore volume of WTS2 as compared to WTS1, which logically also increases the number of available adsorption sites of the former [34]. The better performance of WTS2 may also suggest that a greater portion of AC was available in this sludge for adsorption, and that therefore the reactivation processes may not even be necessary. However, other adsorption mechanisms not evaluated in the present study could also be responsible for the differential interactions between the EPs and the two WTSs evaluated, such as π–π interactions, hydrogen bonding or electrostatic interactions, and this should be explored in future studies [67]. Short-term future prospects for this research are to complete the full characterisation of WTS, including complete textural characterisation and surface chemistry evaluation to identify surface functional groups by Fourier-transform infrared (FT-IR) and X-ray diffraction (XRD) analysis, and also to perform kinetic and equilibrium assays to elucidate the adsorption mechanisms. The pH and temperature dependence of the adsorption process will also be assessed, along with analysis of the stability of the WTS to ensure the safe application of this material. Long-term prospects for this research will include WTS adsorbent capacity analysis by using a real wastewater matrix in order to analyse the removal efficiency of both E2 and EE2 with the aim of bringing the research closer to a real situation.

## 3. Materials and Methods

### 3.1. Raw Material

Two WTSs from different WTPs in Portugal were used to evaluate their adsorption potential of E2 and EE2: (i) Santa Águeda WTP in the Castelo Branco region (WTS1) and (ii) Caldeirão WTP in the Guarda region (WTS2). Both these WTPs use powdered AC in their water treatment process to remove flavour and odours from raw water. Therefore, the sludge produced in the sedimentation tanks of these WTPs will always contain AC. Sludges were collected from these WTPs after the final stage of dewatering by filter press, and the collected sludge was kept in the sun for one month to completely remove the remaining water. The obtained material was not subjected to a reactivation process, and the dried material was ground and sieved to obtain a particle size of 45/60 mesh (250–354 µm), between conventional PAC and GAC particle sizes [68,69,70].

### 3.2. Analytical Methods

#### 3.2.1. WTS Characterisation

Elemental analyses (quantification of carbon, hydrogen, nitrogen and sulphur contents) were performed using an Elemental Thermo Finnigan Analyzer—CE Instruments, model Flash EA 1112 CHNS series (Waltham, MA, USA), based on sample combustion dynamics. The determination of the ash content followed the ASTM D 1762-84 guideline (750 °C) [71]. The pH at the point of zero charge (pHpzc) determination was performed according to the methodology presented by Bernardo et al. [59]. Thermogravimetric analysis (TGA) was performed with Setaram Labsys EVO equipment (Caluire, France) between room temperature and 900 °C with a heating rate of 5 °C/min under argon atmosphere. The mineral analysis was performed by inductively coupled plasma atomic emission spectroscopy (ICP-AES) (Horiba Jobin-Yvon equipment, Kyoto, Japan) after acidic digestion of the WTS samples for the quantification of the following elements: Al, As, Ca, Cd, Cr, Cu, Fe, Hg, K, Mg, Mn, Mo, Na, Ni, P, Pb, Sb, Se and Zn. Textural parameters such as surface area and total pore volume (V_total_) were evaluated from the adsorption of N_2_ at 77 K (ASAP 2010 Micromeritics equipment, Atlanta, GA, USA) by using the single point method at the relative pressure of p/p_0_ = 0.3.

#### 3.2.2. Stock Solution and Determination of the Emerging Pollutants

Stock solutions for both 17β-estradiol (E2; Acros Organics, 98% purity, China) and 17α-ethinylestradiol (EE2; Dr. Ehrenstorfer, 97% purity, Germany) were prepared with a concentration of 500 µg/L for each hormone. The stock solutions were stored in a fridge at 4 °C. Each WTS was added to three separate solutions with different hormone concentrations (200, 350 and 500 ng/L) by using a solid/liquid ratio of 5 g/L. The solutions with WTS were submitted to agitation (200 rpm) in jar test equipment for 24 h. After mixing, the samples were filtered through glass microfibre filters (1.2 µm, GF/C—WATERS) under vacuum. The extraction and detection of the studied EPs were performed by solid-phase extraction (SPE) and high-performance liquid chromatography tandem mass spectrometry (HPLC-MS-MS), according to Gaffney et al. [72,73].

## 4. Conclusions

The results obtained indicate that the tested WTSs present high adsorption potential for both E2 and EE2. Without regeneration or any kind of modification, this adsorbent allowed the achievement of a considerably high, up to a complete, removal of these hormones. These promising results indicate the potential of WTS without addition or activation of AC (i.e., with only AC present from the regular WTP process) in the removal of EPs and may lead to new ways of transforming this erstwhile residue into a possible value-added product. Further studies should be conducted to fully characterise these adsorbent materials through a complete textural characterisation and surface chemistry evaluation to identify surface functional groups and also to perform kinetic and equilibrium assays to elucidate the adsorption mechanisms.

## Figures and Tables

**Figure 1 molecules-26-01010-f001:**
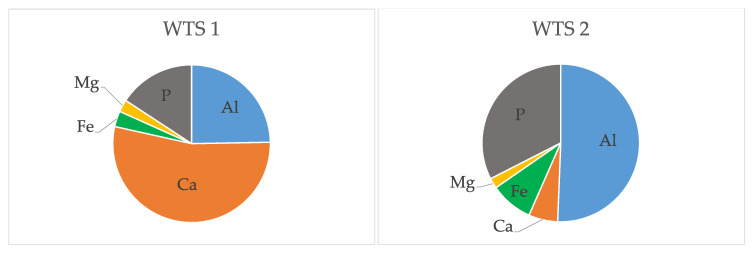
Mineral composition of drinking water treatment sludge (WTS) samples in *w*/*w*% for the major elements.

**Figure 2 molecules-26-01010-f002:**
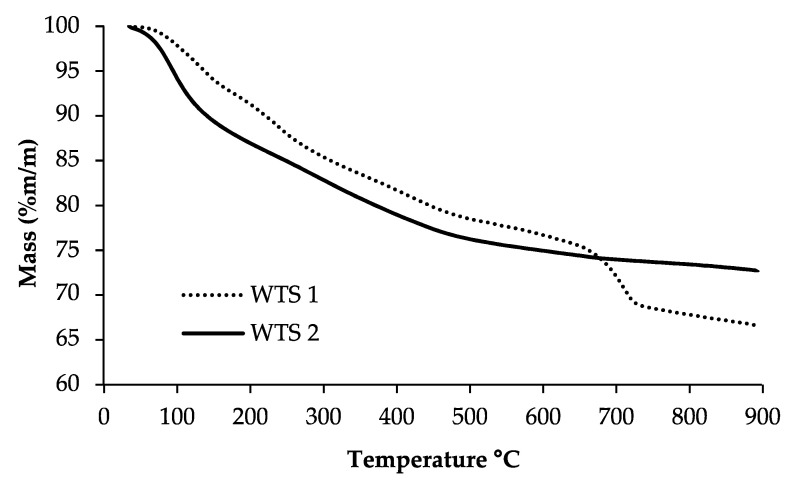
Thermogravimetric analysis (TGA) curves for the two water treatment sludges (WTSs) evaluated.

**Table 1 molecules-26-01010-t001:** Elemental analysis, ash content, pH at the point of zero charge (pHpzc) and textural parameters of the selected drinking water treatment sludges (WTSs). The values presented for the ash content correspond to the mean of duplicates ($\bar{X}\pm \sigma$, *n* = 2).

Parameter	WTS1	WTS2
**C (*w*/*w*%)**	24.31	34.09
**N (*w*/*w*%)**	0.24	0.46
**H (*w*/*w*%)**	2.64	2.92
**S (*w*/*w*%)**	0.16	0.18
**O (*w*/*w*%) ***	29.78	31.49
**Ash (*w*/*w*%)**	42.9 ± 0.2	30.8 ± 0.8
**Surface area (m^2^/g)**	127	318
**Vtotal (cm^3^/g)**	0.065	0.161
**pHpzc**	11.29	7.46

* Obtained by difference (%O = 100 − %C − %H − %N − %S − %Ashes).

**Table 2 molecules-26-01010-t002:** Mineral composition of drinking water treatment sludge (WTS) samples. The values presented correspond to the mean of replicates ($\bar{X}\pm \sigma$, *n* = 2)

Mineral Composition (mg/kg)	WTS1	WTS2
**Al**	53,000 ± 800	54,450 ± 5 350
**As**	37.4 ± 0.9	65.6 ± 7.9
**Ca**	115,350 ± 1650	6429.54 ± 733.41
**Cd**	n.d.	n.d.
**Cr**	n.d.	n.d.
**Cu**	4.1 ± 0.4	10.5 ± 1.1
**Fe**	6890.5 ± 502.9	9442.5 ± 1328.5
**Hg**	n.d.	n.d.
**K**	275.8 ± 10.7	119.9 ± 13.0
**Mg**	5424.0 ± 112.0	2280.3 ± 458.7
**Mn**	468.6 ± 19.0	150.4 ± 14.1
**Mo**	5.3 ± 0.1	5.2 ± 0.6
**Na**	16.1 ± 0.3	5.4 ± 0.5
**Ni**	n.d.	n.d.
**P**	33,817.6 ± 588.5	34,947.3 ± 4242.3
**Pb**	13.1 ± 0.2	14.1 ± 2.6
**Sb**	37.0 ± 0.6	37.4 ± 3.8
**Se**	38.4 ± 0.2	41.3 ± 3.8
**Zn**	21.8 ± 1.4	15.5 ± 1.4

n.d.—not detected.

**Table 3 molecules-26-01010-t003:** Final concentrations of the emerging pollutants in the two water treatment sludges (WTSs) evaluated.

Compound	Initial Concentration (ng/L)	WTS1	WTS2
		Final Concentration (ng/L)	Final Concentration (ng/L)
E2	600 ± 200 (500 ^(2)^)	60 ± 20	<LOQ ^(1)^
	400 ± 100 (350 ^(2)^)	80 ± 30	<LOQ ^(1)^
	300 ± 100 (200 ^(2)^)	<LOQ ^(1)^	<LOQ ^(1)^
EE2	600 ± 200 (500 ^(2)^)	150 ± 60	<LOQ ^(1)^
	400 ± 100 (350 ^(2)^)	170 ± 70	<LOQ ^(1)^
	300 ± 100 (200 ^(2)^)	100 ± 40	<LOQ ^(1)^

^(1)^ LOQ value—50 ng/L; ^(2)^ Initial target concentration.

## Data Availability

The data presented in this study are available on request from the corresponding author.

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
