# Peer review of "Study of the Potential of Water Treatment Sludges in the Removal of Emerging Pollutants"

_molecules, 2021, doi:10.3390/molecules26041010_

Round 1

Reviewer 1 Report

Clearly written article. Both the purpose and why these two hormones were selected for the study were explained.

I understand that the authors intend to continue the interesting research that has been started. I would suggest looking at how pH and temperature affect hormone removal. It is also worth checking the stability of the WTS (pH, time, temperature).

One important note about the article:
Please complete the standard deviations in Table 1, since the authors provide n = 2.

Reviewer 2 Report

Review of the article "Study of the potential of water treatment sludges in the removal of emerging pollutants" by Rita Dias et al. a well-written article. The following points must be considered before accepting this review:

  1. The introduction must be improved by addressing the key points of this research front.
  2. Please explain the role of the hormone in wastewater
  3. Please add some characteristics analysis of selected WTS such as FT-IR, XRD, etc.
  4. Please add future prospective on this research topic.
  5. The writing English must be significantly improved. The authors need to ask for some help from a native speaker to go through this short communication.

Round 2

Reviewer 2 Report

I believe the manuscript has been significantly improved and accept in its present form.

This manuscript is a resubmission of an earlier submission. The following is a list of the peer review reports and author responses from that submission.

Round 1

Reviewer 1 Report

The manuscript (molecules-1051053) presents the potential of water treatment sludges in the removal of emerging pollutants. There are some good results in this work, however, there is no novelty in the present work. The reviewer thinks that the manuscript should be rejected for publication in Molecules. 

Reviewer 2 Report

  1. Although the adsorption process with activated carbon (AC) considered being one of the most promising treatment processes with high EP removal capacity but this short communication fails to address the key points of this research front. There should be put short discussion of detailed critical analysis of authors' findings rather than give some references.
  2. Writing English must be significantly improved. Authors need to ask for some help from a native speaker to go through this short communication.

Reviewer 3 Report

Review of the article "Study of the potential of water treatment sludges in the removal of emerging pollutants" by Rita Dias et al.

Pretty well written article. An extensive bibliography is an undoubted advantage. The introduction perfectly presents the subject of the article to the reader. The article is of the "hot-topic" type, as references to the use of WTS including AC are from August 2020 by Lee et al. (ref. [54]).

Editorial and substantive notes:
1) The abbreviation LOQ appears in the abstract and in the article - I have not found its development. Please include.
2) Why were the 2 hormones EE and EE2 selected? I understand the excuse on lines 52-54 that they are problematic. Any other particular reason? Where do they come from in the wastewater?
3) Please correct the numbers and percentages throughout the article - it should be e.g. 42.9% (without spaces) and not 42.9 % (with spaces) (e.g. lines 94, 96 and others).
4) Please remove hyphens (-) in table descriptions, eg Table 1. - Elemental ...; Table 2. - Mineral ...; Table 3. - Final ....
5) In Table 1, please improve the notation of m2/g and cm3/g - the numbers should be superscript.
6) I am asking for the signature of the figures to be unique - sometimes they are Figure 1 and sometimes Figure 2.
7) Line 132 - I understand 34% and 27% weight loss is at end temperature of 900ºC?
8) Line 140 and 144 - et al., please write in italics, like in the rest of the article
9) Line 141 - A concentration of 5g/L was used. Concentrations "20, 5 and 4 g L" are given as a literature explanation. Very unclear for me the record why such. I find it inappropriate to rely on another article (ref. [54]). Especially that in the quoted article [54] I do not find such values. If I am wrong, please correct me.
10) Table 3 - Baseline concentrations 600 +/- 200; 400 +/- 100; 300 +/- 100 and further. Why such large deviations up to 1/3 of the value? It totally undermines the whole experiment, the more so as, as I understand it, these concentrations were prepared by yourself. Why with EE2-there are two times the concentration of 300 +/- 100?
11) Lines 174-176 and 186-187 - it is a pity that the announcements of future research have not been included in this article - it would have increased its value, but I respect the authors' decision.
12) Lini 198 - please unify the number of footnotes [67-69].
13) Paragraph 3.2 - please add the cities and countries of the equipment manufacturers and the origin of E2 and EE2 hormones (producer, purity, city, country).
14) Line 216 - please write Celsius degrees accordingly.
15) In the article, I miss the textural characteristics and pHpzc of the selected WTS particle size distribution (μm).
16) There is also no FT-IR and XRD analysis of the selected WTS.
17) Why was no additional research done, how would the removability of the horomnes change depending on the pH?
18) I understand the absorption studies were done at room temperature. Did the authors wonder if higher temperature would alter hormone absorption? If so, how and why such research was not undertaken?
19) What about the stability of the WST (pH, temperature, time)?